# Anticancer Properties of 3-Dietoxyphosphorylfuroquinoline-4,9-dione

**DOI:** 10.3390/molecules28073128

**Published:** 2023-03-31

**Authors:** Joanna Drogosz-Stachowicz, Katarzyna Gach-Janczak, Marek Mirowski, Jacek Pietrzak, Tomasz Janecki, Anna Janecka

**Affiliations:** 1Department of Biomolecular Chemistry, Medical University of Lodz, Mazowiecka 6/8, 92-215 Lodz, Poland; 2Laboratory of Molecular Diagnostics and Pharmacogenomics, Department of Pharmaceutical Biochemistry and Molecular Diagnostics, Medical University of Lodz, Muszynskiego 1, 90-151 Lodz, Poland; 3Institute of Organic Chemistry, Lodz University of Technology, Żeromskiego 116, 90-924 Lodz, Poland

**Keywords:** quinolinediones, cytotoxic activity, apoptosis, DNA damage, maximum tolerated dose

## Abstract

Herein, the antitumor activity of a novel synthetic analog with 5,8-quinolinedione scaffold, diethyl (2-(2-chlorophenyl)-4,9-dioxo-4,9-dihydrofuro [3,2-*g*]quinolin-3-yl)phosphonate (**AJ-418**) was investigated on two breast cancer cell lines. This analog was selected from a small library of synthetic quinolinediones on the basis of its strong antiproliferative activity against MCF-7 and MDA-MB-231 cells and 4-5-fold lower cytotoxicity towards healthy MCF-10A cells. The morphology of MCF-7 and MDA-MB-231 cancer cells treated with **AJ-418** changed drastically, while non-tumorigenic MCF-10A cells remained unaffected. In MCF-7 cells, after 24 h incubation, the increased number of apoptotic cells coincided with a decrease in proliferation and cell viability. The 24 h treatment of MDA-MB-231 cells with the tested compound reduced their cell viability and proliferation rate; however, a significant pro-apoptotic effect was visible only after longer incubation times (48 h and 72 h). Then, the maximum tolerated dose (MTD) of compound **AJ-418** in C3H mice after subcutaneous administration was determined to be 160 mg/kg, showing that this analog was well tolerated and can be further evaluated to assess its potential therapeutic effect in tumor-bearing mice.

## 1. Introduction

Heterocyclic compounds play an important role in the development of anticancer agents [1] because the ability of heteroatoms to participate in hydrogen bonding with the target protein can improve the selectivity and potency of the molecules [2]. The scaffold of quinolinediones is often found in nature and used for the development of new, biologically active substances. Among many possible isomeric forms, the most common are 5,8-quinolinediones (Figure 1), which are present in several structures of natural and synthetic compounds [3]. Two naturally occurring antitumor antibiotics based on 5,8-quinolinedione structure are streptonigrin [4] and lavendamycin [5] (Figure 1). Streptonigrin was found to be a potent anti-leukemia agent. However, a high degree of toxicity caused the termination of its development at phase II clinical trials [6]. The use of lavendamycin in cancer therapy failed, due to poor water solubility and low specificity of this compound. The study of lavendamycin-based compounds designed to overcome these drawbacks has been an area of research [7].

Several synthetic compounds containing 5,8-quinolinedione moiety that exhibit anticancer activity have been described in the literature. Ling and co-workers obtained a series of 6- and 7-arylamino-5,8-quinolinediones that were tested against drug sensitive (HeLeS3) and multidrug resistant (KB-vin) cell lines [8]. Other mono and disubstituted derivatives of the 5,8-quinolinedione were synthesized and showed high cytotoxic activity against several tumor cell lines [9,10,11]. More recently, a series of diversely substituted furoquinolinediones has been synthesized and some of these analogs turned out to be inhibitors of tyrosyl-DNA phosphodiesterase-2 (TDP2), the enzyme that repairs topoisomerase II (TOP2)-mediated DNA damage [12].

On the other hand, heterocycles containing a phosphorus substituent have received a lot of attention mainly due to their applications in chemical synthesis and their increasing usefulness in drug design [13]. A number of phosphorus-containing drugs have been designed as prodrugs in the form of phosphotriesters, phosphonates, phosphinates and phosphorus oxides to reduce side effects and toxicity and achieve greater selectivity and bioavailability. Such prodrugs are believed to have a higher polarity and enable the formation of stronger hydrogen bonds in vivo compared to unmodified molecules [14]. Several important drugs have been designed on this basis, fludarabine phosphate with anticancer activity being one example [15]. In addition, it has been shown that the presence of a dialkoxy- or diaryloxyphosphoryl group can improve the bioavailability of drugs [16], as well as slow down the progression of cancer by inhibiting, among others, farnesyl transferases [17] or purine nucleoside phosphorylases [18].

In our previous paper [19] we reported the synthesis of 3-diethoxyphosphorylfuroquinoline-4,9-diones combining furoquinoline-5,8-dione skeleton with diethoxyphosphoryl moiety. All obtained compounds proved to be highly cytotoxic for breast cancer adenocarcinoma MCF-7 and promyelocytic leukemia HL-60 cells, with IC_50_ values in the low µM range. The mode of action of the most potent analogs was investigated in HL-60 cells. These analogs enhanced intracellular ROS generation and NAD(P)H:quinone oxidoreductase 1 (NQO1) depletion, which led to reduced cell proliferation, DNA damage and apoptosis.

In the extension of this research, herein, we investigated the influence of diethyl (2-(2-chlorophenyl)-4,9-dioxo-4,9-dihydrofuro[3,2-*g*]quinolin-3-yl)phosphonate (**AJ-418**) (Figure 2) on cancerous mammary cell lines, MCF-7 and MDA-MB-231, in comparison with the non-tumorigenic MCF-10A cells. Then, the maximum tolerated dose (MTD) of the new analog in C3H mice was assessed in vivo.

## 2. Results

### 2.1. In Vitro Data

#### 2.1.1. Cytotoxic Activity against Breast Cancer and Non-Tumorigenic Cell Lines (MTT Assay)

The cytotoxicity of **AJ-418** for MCF-7, MDA-MB-231 and non-tumorigenic MCF-10A cells was assessed after 48 h and 24 h incubation in the concentration range of 0–2.5 µM using an MTT assay. The obtained IC_50_ values (Table 1) were similar for both cancer cell lines and 4- to 5-fold higher for the MCF-10A non-tumorigenic breast cell line, indicating that the tested compound showed some selectivity of action.

#### 2.1.2. Morphology of MCF-7, MDA-MB-231 and MCF-10A Cells Incubated with AJ-418

Morphology changes in MCF-7 and MDA-MB-231 cancer cells, and for comparison in non-tumorigenic MCF-10A breast cells, were assessed after staining these cells with Giemsa dye that stains nuclei dark blue and cytoplasm blue or pink. The control cells (untreated) grew well and exhibited epithelial features, forming a monolayer. The morphology of MCF-7 and MDA-MB-231 cells treated with **AJ-418** for 24 h changed, the cells became more heterogeneous in shape and size, and a reduction in cell number was observed. Importantly, the morphology of non-tumorigenic MCF-10A breast cells treated with the tested compound at either 0.5 µM or 1 µM concentrations did not change compared to the control (Figure 3).

#### 2.1.3. Apoptosis Induction

Apart from morphological changes, the typical features of apoptotic cell death are phosphatidylserine (PS) translocation to extracellular membrane, caspase-3 activation and DNA damage.

The translocation of PS in MCF-7 and MDA-MB-231 cells treated with **AJ-418** was assessed using double-staining with Annexin V and propidium iodide (PI). As shown in Figure 4, after 24 h, **AJ-418** did not significantly increase the number of MCF-7 cells in the early phase of apoptosis, while increased late apoptosis events from 14.9 ± 0.57% in control (untreated) cells to 47.7 ± 0.6% and 60.8 ± 1.4% for 0.5 and 1 µM concentrations, respectively. In MDA-MB-231 cells, 24 h incubation with **AJ-418** induced early apoptosis only in a small percent of cell population. Longer incubation (48 h and 72 h) led to the induction of both early and late apoptosis, which was more pronounced when a higher concentration of **AJ-418** (1 µM) was used. At this concentration, the tested compound enhanced the percentage of apoptotic cells to 7.1 ± 0.21% and 6.7 ± 0.16%, for early apoptotic occurrence and to 16.3 ± 0.83% and 39.9 ± 0.44% for late apoptosis, after 48 h and 72 h incubation, respectively. Thus, the pro-apoptotic effect of compound **AJ-418** in MDA-MB-231 cells was dose and time dependent (Figure 4).

The induction of apoptosis was also evaluated by detecting a cleaved fragment of PARP (poly(ADP-ribose) polymerase) with a mass of 89 kDa, generated by cleavage of PARP1 by caspase 3 and 7 during caspase-mediated apoptosis [20]. **AJ-418** at 0.5 and 1 µM concentrations dose-dependently increased the population of cleaved PARP in MCF-7 cells to 28.7 ± 2.95% and 36.2 ± 0.64%, respectively, compared to controls (2.7 ± 0.52%). In MDA-MB-231 cells, these values were lower, 1.6 ± 0.18% and 1.2 ± 0.2% of the cell population, respectively, compared to controls (0.5 ± 0.07%) (Figure 5). The results suggest that after 24 h, **AJ-418** at the tested concentrations caused significant apoptosis in MCF-7 cells, while in MDA-MB-231 cells, only a weak pro-apoptotic effect was observed.

#### 2.1.4. DNA Damage Assessment

Apoptosis induced by many anticancer drugs may be a consequence of DNA damage [21]. Chemical genotoxins that target DNA can inhibit DNA replication, which leads to the collapse of replication forks and the formation of DNA double-strand breaks (DSBs). An early cells’ response to DNA damage includes the histone variant H2AX phosphorylation on Ser139, which leads to structural modifications at the site of damage. Phosphorylated H2AX has been shown to promote DNA repair and play a role in genome stability, cell cycle checkpoint response, and tumor suppression [22]. Therefore, the measurement of histone H2AX phosphorylation level can be used to detect the genotoxic effects of potential anticancer agents.

To assess the ability of **AJ-418** to phosphorylate H2AX, MCF-7 and MDA-MB-231, cells were incubated with the tested compound for 24 h at 0.5 and 1 µM concentrations. **AJ-418** caused a significant H2AX phosphorylation, indicative of DNA damage in both MCF-7 and MDA-MB-231 cell lines (Figure 6). **AJ-418** phosphorylated H2AX in 33.2 ± 2.14% and 48.4 ± 0.18% of MCF-7 cells, respectively, compared to controls (2.7 ± 0.03%) and was slightly less genotoxic to MDA-MB-231 cells, 33.0 ± 3.19% and 36.9 ± 2.03%, respectively, compared to controls (0.8 ± 0.13%).

#### 2.1.5. Antiproliferative Activity

The ability of **AJ-418** to inhibit MCF-7 and MDA-MB-231 cell proliferation was evaluated by exposure of cells to 5-bromodeoxyuridine (BrdU), a thymidine analog that incorporates into newly synthesized DNA during the S-phase of the cell cycle. The obtained results showed that the tested compound significantly inhibited the proliferation of both cell lines. Analog **AJ-418** at 0.5 and 1 µM concentrations caused the decrease in BrdU-incorporating MCF-7 cells from 37.9 ± 2.8% in the control to 26.9 ± 2.24% and 7.4 ± 2.02%, respectively, while in MDA-MB-231 cells, **AJ-418** reduced proliferation from 56.0 ± 1.26% in the control to 32.6 ± 0.76% and 22.7 ± 1.86%, respectively (Figure 7).

#### 2.1.6. Cell Cycle Analysis

Cytotoxic action of anticancer drugs may be exerted, among others, by the arrest of the cell cycle. Changes in the distribution of the cell cycle phases after treatment with **AJ-418** were observed using flow cytometry analysis. After 24 h incubation with this compound, MCF-7 and MDA-MB-231 cells were treated with a fluorescent dye (4,6-diamidino-2-phenylindole, DAPI), which quantitatively stains DNA. The fluorescence intensity of the stained cells corresponds to the amount of DNA they contain and indicates the position in the cell cycle in the major phases (G0/G1 vs. S vs. G2/M). Representative histograms showing changes in the cell cycle distribution are presented in Figure 8. **AJ-418** at 0.5 µM and 1 µM concentrations caused a significant increase in the number of MCF-7 cells in the S and G2/M phases (from 18.2% and 12.0% in the control to 27.5% and 19.0%, respectively), and in MDA-MB-231 cells (from 20.1% and 16.9% in the control to 25.4% and 20.3% in treated cells, respectively). At the higher concentration (1 µM), there was a significant increase in the number of MCF-7 cells primarily in the sub G0/G1 phase (from 1.8% in the control to 14.6%). In contrast, in MDA-MB-231 cells, the higher concentration of **AJ-418** did not result in significant changes in the cell cycle distribution compared to controls.

### 2.2. In Vivo Studies—Evaluation of Maximum Tolerated Dose in C3H Mice

The aim of the in vivo studies was to estimate the toxicity of the tested compound in C3H mice. To determine the maximum tolerated dose (MTD) of compound **AJ-418** in C3H mice after subcutaneous (s.c.) administration, on the first day of the experiment, a group of three mice was given an initial dose of the compound, 20 mg/kg, after which the animals were observed for 48 h. During this period no signs of compound toxicity, including weight loss, changes in appearance, body posture, mobility or changes at the site of administration were observed. Then, the other four groups of mice (n = 3) were administered sequentially higher doses of the compound, 40 mg/kg, 80 mg/kg, 160 mg/kg and 240 mg/kg. Animals were observed for 48 h between doses and no adverse effects were detected. Within an hour after administration of a dose of 240 mg/kg, no negative symptoms were seen in the animals that could suggest a toxic effect of the tested compound. However, the next morning all mice (n = 3) were found dead. Therefore, the penultimate dose of 160 mg/kg administered s.c. was considered the MTD of compound **AJ-418** in C3H mice (Figure 9).

## 3. Discussion

Cancer tumors often consist of different types of cells; therefore, the phenotype of cancer cells is commonly considered an important parameter to determine the degree of their malignancy, and consequently, the response that can be expected from the applied therapy [23]. Here, the effect of **AJ-418** on cell viability and proliferation was analyzed on two human breast cancer cell lines, MCF-7 and MDA-MB-231, with different proliferative potential.

MCF-7 cells express estrogen receptors (ER), progesterone receptors (PR) and epidermal growth factor receptors (EGFR). It was shown that estrogens stimulate the proliferation of breast cancer cells in vitro and in vivo [24,25]. ER activation is considered an important factor in breast cancer progression [26], whereas treatment with anti-estrogen chemotherapy drugs (e.g., tamoxifen) can reduce the growth of cultures by inhibiting proliferation and inducing apoptosis [27].

The MDA-MB-231 is defined as a triple-negative breast cancer (TNBC) cell line, as it lacks ER, PR and EGFR. This cell line is commonly used to model late-stage breast cancer [28]. TNBC is characterized by high invasiveness, high metastatic potential and short-term response to available therapies, with the tendency to relapse and with poor prognosis [29]. Since TNBC tumors do not express ER, PR, and EGFR, they are not sensitive to endocrine therapy or EGF therapy, and standardized TNBC treatment regimens are not available. Therefore, the development of new strategies for the treatment of TNBC continues to be the goal of many studies [30,31].

The human non-tumorigenic breast cell line MCF-10A is widely used for comparison in various studies, due to its structural similarity to the normal human mammary epithelium. MCF-10A cells are controlled by hormones and growth factors, although they do not express ER or show signs of terminal differentiation or senescence [32]. In mice, these cells were found to be non-tumorigenic [33].

In this report, it was shown that **AJ-418** was cytotoxic to MCF-7 and MDA-MB-231 cells in a concentration-dependent manner but had no apparent effect on the growth of MCF-10A cells, nor did it induce noticeable changes in their morphology. The increase in the number of apoptotic MCF-7 cells observed after incubation with **AJ-418** coincided with a decrease in proliferation and cell viability. In MCF-7, but not in MDA-MB-231 cells, **AJ-418** caused a significant increase in the number of cells in the sub-G0/G1 phase, with reduced DNA content, indicating their programmed death. In MDA-MB-231 cells, a significant pro-apoptotic effect was visible only after a longer incubation time (72 h). This observation is in agreement with the fact that the MDA-MB-231 cell line represents one of the most aggressive, therapy-resistant and metastatic cancers.

The cytotoxic effect of anticancer agents often correlates with DNA damage and double-strand breaks (DSBs) [21]. **AJ-418** led to a significant increase in DSBs in almost half of the MCF-7 cell population and was slightly less genotoxic to MDA-MB-231 cells. The exposure of MCF-7 cells to the tested compound led to an increase in γH2AX, leading to DNA damage and apoptosis. Interestingly, in MDA-MB-231 cells, despite a similar level of DNA damage, apoptosis occurred in a smaller number of cells, which probably indicates that DSBs could be repaired and did not immediately lead to cell death. These results show that similar to other anticancer drugs, **AJ-418** exerts a stronger therapeutic effect on MCF-7 cells, while higher efficiency of DNA repair in MDA-MB-231 cells may be one of the factors contributing to their chemoresistance [34].

In order to determine the in vivo toxicity of **AJ-418**, this analog was administered to C3H mice. The C3H strain was obtained by Strong in 1920 [35]. Female C3H mice that are neonatally infected with the murine breast cancer virus develop tumors spontaneously and with high frequency [36]. The MTD of **AJ-418** in C3H mice after a single s.c. administration, was determined to be 160 mg/kg b.w., indicating that this analog was well tolerated.

In summary, **AJ-418** was cytotoxic to MCF-7 and MDA-MB-231 tumor cells, but had no apparent effect on the growth of non-tumorigenic MCF-10A cells. In MCF-7 cells, the tested analog induced the formation of DSBs, which led to the cell cycle arrest in the S and G2/M phases, and finally to apoptosis. In MDA-MB-231 cells, **AJ-418** increased the number of cells with DSBs, but significant differences in the cell cycle distribution were not observed and apoptosis occurred after longer incubation times, indicating that this cell line was more resistant than MCF-7 to the action of **AJ-418**. In vivo, the new analog was well tolerated in female C3H mice (Figure 10).

These findings indicate that synthetic 3-dietoxyphosphorylfuroquinoline-4,9-dione **AJ-418** can be further modified and studied in vivo as a potential anticancer agent for breast cancer therapy.

## 4. Materials and Methods

### 4.1. Cell Cultures and Sample Preparation

Breast cancer MCF-7 and MDA-MB-231 cell lines were purchased from the European Collection of Cell Cultures (ECACC) and cultured in Minimum Essential Medium Eagle (MEME, Sigma–Aldrich, St. Louis, MO, USA) supplemented with 10% heat-inactivated fetal bovine serum (FBS) (Biological Industries, Beit-Haemek, Israel), 2 mM glutamine, Men Non-essential amino acid solution and antibiotics (100 μg/mL streptomycin and 100 U/mL penicillin), all from Sigma–Aldrich (Sigma–Aldrich, St. Louis, MO, USA). MCF-10A were purchased from the American Type Culture Collection (ATCC) and cultured using MEGM Mammary Epithelial BulletKit. Cells were maintained at 37 °C in a 5% CO_2_ atmosphere and grown until 80% confluent.

For biological experiments, the tested compound was dissolved in DMSO (Sigma–Aldrich, Louis, MO, USA) and further diluted in culture medium to obtain less than 0.1% DMSO concentration. In each experiment, controls without and with DMSO were performed. DMSO in 0.1% concentration had no effect on the observed parameters.

### 4.2. MTT Assay

The MTT (3-(4,5-dimethyldiazol-2-yl)-2,5 diphenyl tetrazolium bromide) assay was performed according to Mosmann [37]. The cells were incubated with the tested compound for 24 h or 48 h. The absorbance was measured at 560 nm using FlexStation 3 Multi-Mode Microplate Reader (Molecular Devices, LLC, San Jose, CA, USA) and compared with control (untreated cells). Experiments were performed in triplicate. The IC_50_ values were calculated from concentration–response curves.

### 4.3. Assessment of Morphological Changes

Cells (MCF- 7, MDA-MB-231, MCF-10A) were seeded in 6-well plates at 2 × 10^5^ cells/well in 2 mL of culture medium and incubated for 24 h. After another 24 h incubation with the tested compound, the old medium was removed and the cells were stained by adding 1 mL of Giemsa reagent (Sigma–Aldrich, St. Louis, MO, USA) at a 1:2 dilution in deionized water. The plates were incubated at 37 °C for 2 h and then rinsed thoroughly with distilled water. Morphological changes in cells treated with the tested compound were observed and photographed using a digital camera (MoticMoticam 2300, Motic China Group Co.,Ltd., Xiamen, China) connected to a phase-contrast microscope.

### 4.4. Apoptosis Detection by Annexin V and Propidium Iodide Staining

The induction of apoptosis was determined using a FITC Annexin V Apoptosis Detection Kit I (BD Biosciences, San Jose, CA, USA). The percentage of apoptotic cells was assessed using flow cytometry with CytoFLEX (Beckman Coulter, Inc., Brea, CA, USA). Data analysis was performed using Kaluza Analysis Software v2 (Beckman Coulter, Inc., Brea, CA, USA).

### 4.5. Evaluation of Cell Cycle Distribution, Proliferation, Apoptosis and DNA Damage Using the “Apoptosis, DNA Damage and Cell Proliferation Kit”

Cell cycle kinetics, apoptosis and DNA damage were assessed using the “Apoptosis, DNA Damage and Cell Proliferation Kit” (BD Biosciences, San Jose, CA, USA) according to the manufacturer’s protocol. MCF-7 and MDA-MB-231 cells were seeded into bottles at 2 × 10^5^ cells/mL in 10 mL of medium culture and left for 24 h. Then, the tested compound in the specific concentrations was added. Cells incubated without the tested compound served as controls. After 24 h incubation, the cells were treated with a BrdU solution in a final concentration of 10 µM in culture medium and incubated for 8 h. Cells were trypsinized, collected using centrifugation, fixed, permeabilized and then incubated with DNase (300 µg/mL in PBS) for 1 h at 37 °C to expose the incorporated BrdU. Then, the cells were stained with Anti-BrdU, Anti-H2AX (pS139) and Anti-Cleaved PARP (Asp214) antibodies for 20 min in the dark at room temperature. Finally, the cells were resuspended in DAPI solution (1 µg/mL in staining buffer) to stain total DNA for cell cycle analysis. Cells were analyzed using flow cytometry using CytoFLEX (Beckman Coulter, Inc.). The analysis of data was performed using Kaluza Analysis Software v2 (Beckman Coulter, Inc.).

### 4.6. Animals

Animal care and all experimental procedures were performed in accordance with the Medical University of Lodz recommendations, described in the Guide for the Care and Use of Laboratory Animals and complied with the ARRIVE guidelines [38].

All experiments on living animals were carried out according to protocols approved by the Local Ethical Committee No: 41/ŁB 180/2020 20 July 2020.

Female C3H mice (Department of Genetics and Laboratory Animal Breeding, Oncology Center, Institute of Maria Skłodowska-Curie, Warsaw, Poland), weighing 22–25 g were used in the study. The animals were housed under standard conditions (22 ± 1 °C, 12 h light/dark cycle) with food and water ad libitum for 7 days before the experiments. 

### 4.7. Determination of the Maximum Tolerated Dose (MTD)

Fifteen C3H mice were randomly assigned to groups of n = 3 animals. The first group was given s.c. an initial dose (20 mg/kg). This dose was determined on the basis of bibliographic data on the MTD values of similar cytotoxic drugs in mice [39]. The first dose could be escalated or de-escalated depending on the absence or presence of severe toxicity. Subsequent groups were treated with increasing doses of the tested compound. If none of the three animals in a group developed dose-limiting toxicity (DLT), the next three animals were given the next higher dose. However, if one of the three mice developed dose-limiting toxicity, the next three animals would be given the compound at the same dose level. Dose escalation continued until at least two subjects per group experienced dose-limiting toxicity. Following the administration of each dose, mice were observed for 48 h and the following parameters and endpoints were assessed: mortality, clinical signs and body weight. The MTD in this study was defined as the highest dose that will be tolerated and will not cause serious life-threatening toxicity in an animal for the duration of the study. The MTD was set as the dose preceding the toxic dose.

The solution of the tested compound was administered with a 25-gauge needle in a volume of 0.1 mL/10g b.w. The injection was performed s.c. in the interscapular area.

All surviving animals after the experimental time were euthanized by cervical dislocation. The animals were sacrificed individually, in isolation from other animals, and anesthetized by intraperitoneal injection with a mixture of ketamine (50 mg/kg b.w.) and xylazine (5 mg/kg b.w.).

### 4.8. Statistical Analysis

For the in vitro studies, all data were expressed as mean ± SEM. Statistical analyzes were performed using Prism 6.0 (GraphPad Software Inc., San Diego, CA, USA). Statistical significance was assessed using one-way ANOVA, followed by the Student–Newman–Keuls post-hoc multiple comparison test (for comparisons of three or more groups) or the Student’s *t*-test (for comparisons of two groups). The values * *p* < 0.05, ** *p* < 0.01 and *** *p* < 0.001 were considered statistically significant.

## Figures and Tables

**Figure 1 molecules-28-03128-f001:**
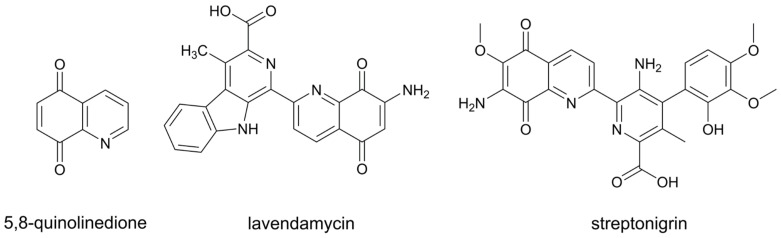
Structures of 5,8-quinolinedione and anticancer antibiotics containing this motif.

**Figure 2 molecules-28-03128-f002:**
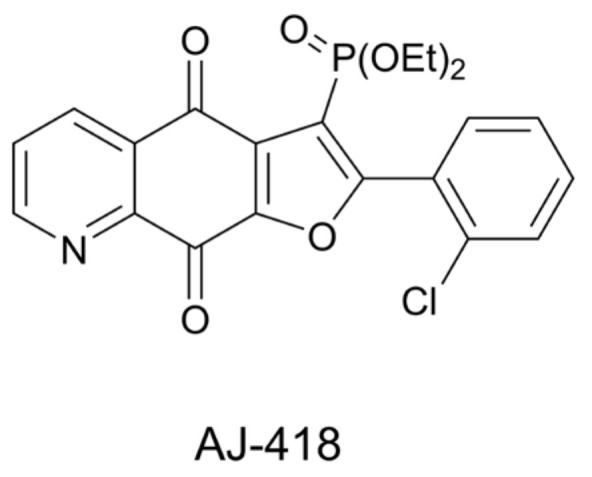
Structure of diethyl (2-(2-chlorophenyl)-4,9-dioxo-4,9-dihydrofuro[3,2-*g*]quinolin-3-yl)phosphonate (**AJ-418**).

**Figure 3 molecules-28-03128-f003:**
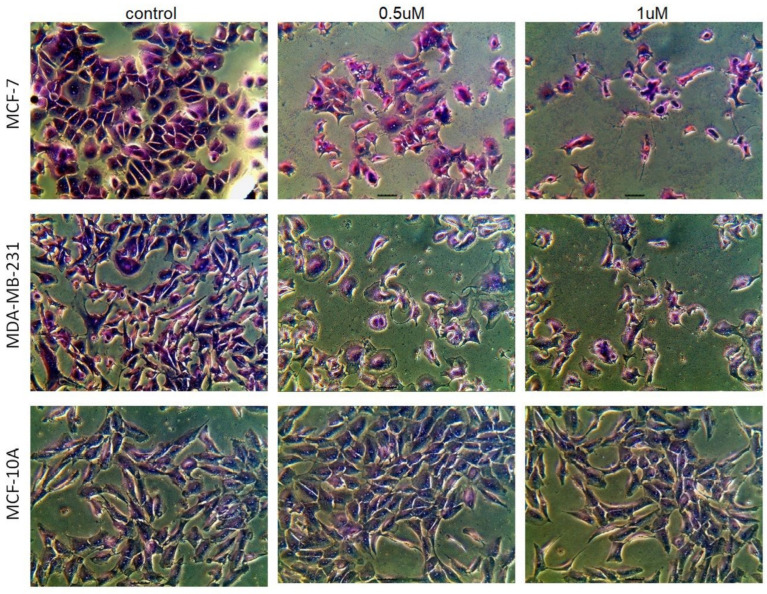
Morphological changes in MCF-7, MDA-MB-231 and MCF-10A cells after 24 h incubation with **AJ-418** at concentrations of 0.5 µM and 1 µM. After staining with Giemsa dye, the cells were photographed under a light microscope with a built-in camera (magnification 40×).

**Figure 4 molecules-28-03128-f004:**
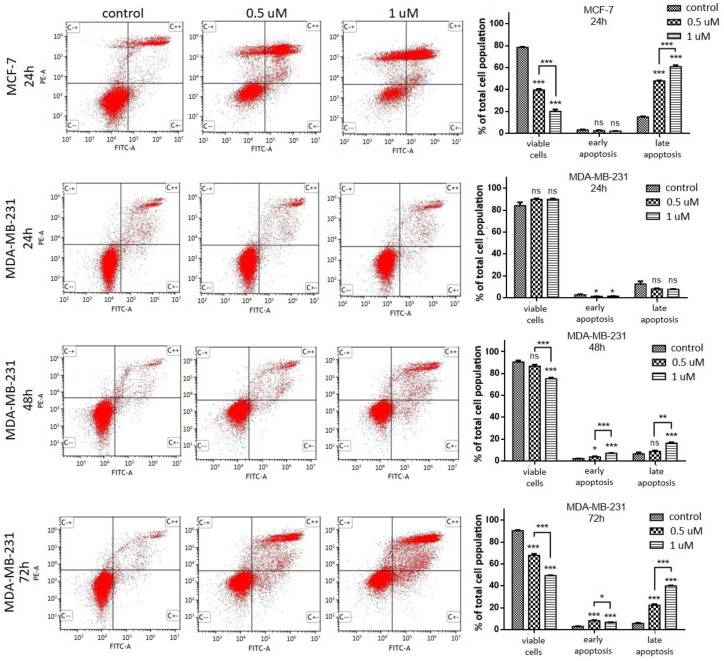
Induction of apoptosis by **AJ-418** in MCF-7 and MDA-MB-231 cells. Cells were exposed to the tested compound at 0.5 µM and 1 µM concentrations, MCF-7 for 24 h and in MDA-MB-231 for 24, 48 and 72 h, then stained with annexin V and PI and analyzed using flow cytometry. Representative cytograms and quantification of apoptotic cell death are shown in the figure; each column represents the mean ± SEM of three independent experiments; * *p* < 0.05, ** *p* < 0.01 and *** *p* < 0.001 were considered statistically significant compared to controls, ns—no statistical significance.

**Figure 5 molecules-28-03128-f005:**
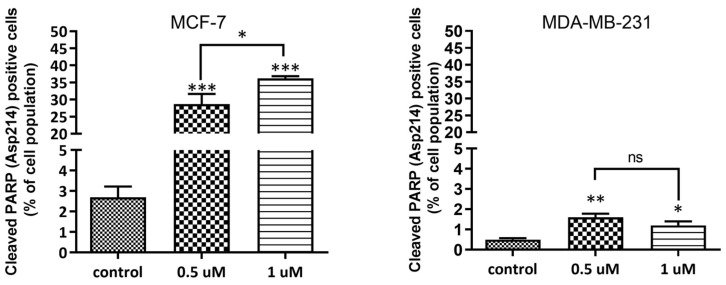
Percentage of the cell population containing cleaved PARP. MCF-7 and MDA-MB-231 cells were exposed to **AJ-418** at 0.5 µM and 1 µM for 24 h, then labeled with Anti-Cleaved-PARP antibody (Asp214) and analyzed using flow cytometry. Data represent means ± SEM of three independent experiments. Statistical significance was determined using one-way ANOVA and the Student–Newman–Kleus post-hoc multiple comparison test; *** *p* < 0.001, ** *p* < 0.01, * *p* < 0.05 represent statistically significant differences in comparison to controls, ns—no statistical significance.

**Figure 6 molecules-28-03128-f006:**
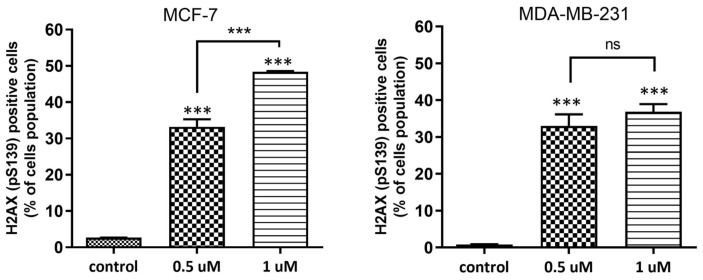
Induction of DNA damage by **AJ-418** in MCF-7 and MDA-MB-231 cells. Cells were exposed to the tested compound at 0.5 µM and 1 µM concentration for 24 h, stained with anti-H2AX antibody (pS139) and analyzed using flow cytometry. Error bars represent means ± SEM of three independent experiments. *** *p* < 0.001 were considered statistically significant compared to controls, ns—no statistical significance.

**Figure 7 molecules-28-03128-f007:**
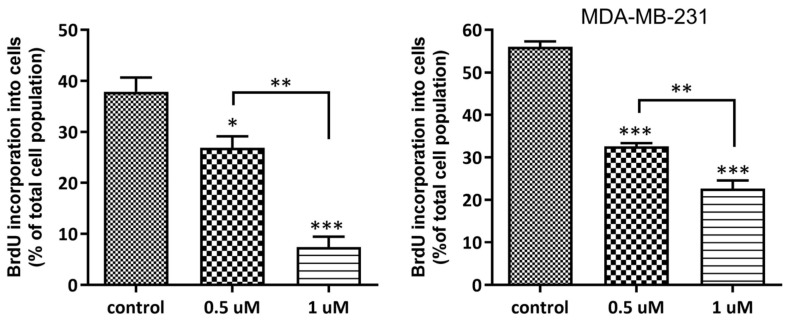
Effect of **AJ-418** on the proliferation of MCF-7 and MDA-MB-231 cells. Cells were exposed to **AJ-418** at 0.5 µM and 1 µM concentrations for 24 h, stained with BrdU and analyzed using flow cytometry. Error bars represent means of three replicates ± SEM. * *p* < 0.05, ** *p* < 0.01 and *** *p* < 0.001 were considered statistically significant compared to controls.

**Figure 8 molecules-28-03128-f008:**
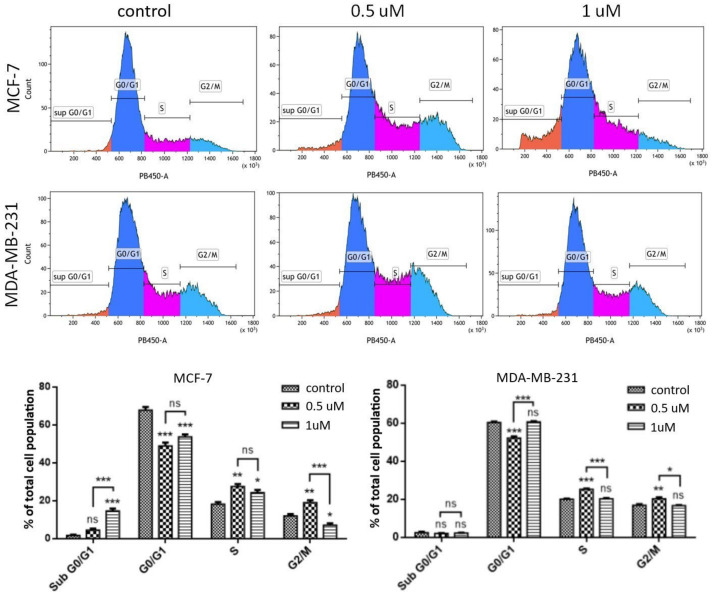
Effect of **AJ-418** on cell cycle distribution in MCF-7 and MDA-MB-231 cells. Cells were exposed to **AJ-418** at 0.5 µM and 1 µM concentrations for 24 h, stained with DAPI and analyzed using flow cytometry. Representative histograms and quantification are shown in the figure; error bars represent means ± SEM of three replicates. * *p* < 0.05, ** *p* < 0.01 and *** *p* < 0.001 were considered statistically significant compared to controls, ns—no statistical significance.

**Figure 9 molecules-28-03128-f009:**
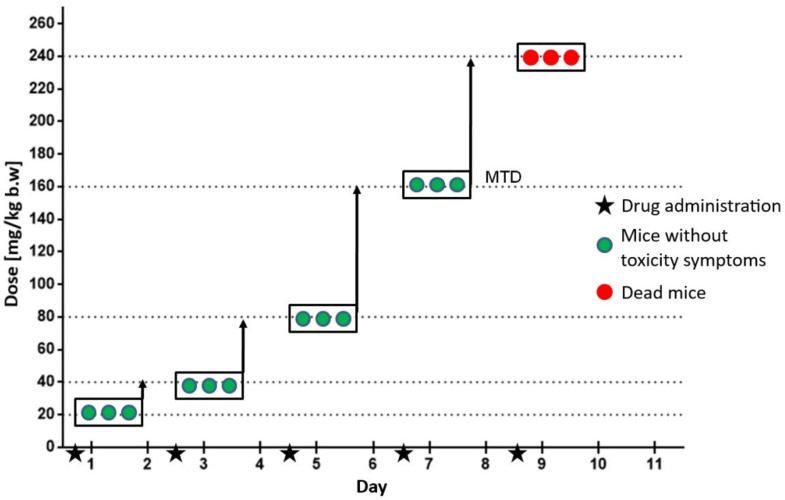
Diagram showing the determination of the maximum tolerated dose (MTD). Each box represents a group of 3 mice that were given the tested compound at the specific dose. After each administration, the mice were observed for 48 h.

**Figure 10 molecules-28-03128-f010:**
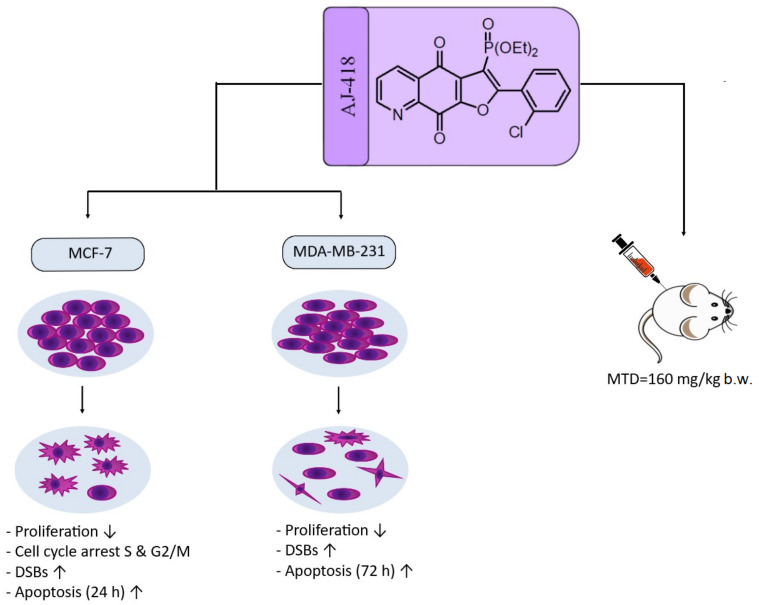
Schematic representation of the effect of quinolinedione **AJ-418** on MCF-7 and MDA-231 cells and evaluation of MTD in C3H mice.

**Table 1 molecules-28-03128-t001:** Cell viability of MCF-7, MDA-MB-231 and MCF-10A cells after 48 h and 24 h incubation with **AJ-418**, assessed using the MTT test.

Cell Line	IC_50_ (µM) ^a^		
	AJ-418		Doxorubicin
	48 h	24 h	48 h
MCF-7	0.10 ± 0.01	0.46 ± 0.01	0.89 ± 0.12
MDA-MB-231	0.12 ± 0.01	0.49 ± 0.01	0.68 ± 0.08
MCF-10A	0.50 ± 0.01	1.82 ± 0.04	1.2 ± 0.05

^a^ Compound concentration required to inhibit cell viability by 50% after 48 h or 24 h treatment. Data are expressed as the mean ± SEM from the concentration–response curves of at least three experiments.

## Data Availability

Not applicable.

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
