# Peer review of "Anticancer Properties of 3-Dietoxyphosphorylfuroquinoline-4,9-dione"

_molecules, 2023, doi:10.3390/molecules28073128_

Round 1

Reviewer 1 Report

This paper presents the results of well-designed and performed study on the anti-cancer activity of novel synthetic analog of 5,8-quinolinedione. Basically, the paper could be accepted for publication in the present form. However, some minor corrections will improve its value.

1.     In the Abstract and in whole paper the authors describe MCF-10A cells as healthy (abstract line 20) or normal (in the text). It is not quite true, since these cells are immortalized. Therefore, correctly should be described as nontumorigenic/immortalized breast cell line.

22.   Abstract - line 23: …and metabolic activity. Any metabolic activity was evaluated in this study. MTT assay although based on NAD(P)H oxidoreductase evaluates cell viability and cannot be considered as the measure of cell metabolism.

33.  Table 1 Again MTT measures basically cell viability not proliferation (if so, rather increase not inhibition). Correction at the Table’s footnote is suggested.

44.  Subtitle 2.2.3 Histological techniques - Histological analysis sounds better.

It seems that tested compound did not change the histological image. It should be clearly stated or this section removed.

55.  Discussion line 312. The reference# 34 describes the DSBs induced by radiotherapy, not chemotherapeutics. Therefore, would be better to replace it with more adequate.

66.   Discussion paragraph between the lines 336-34, presents rather summary, than conclusions. Thus, should be rewritten as real conclusions  or renamed: In summary.

Reviewer 2 Report

Minor corrections

Which concentrations were used in the MTT assay?

Figure 2, What is the relevance of this figure? What is the justification to show a morphological change?

Major corrections

There are no positive controls in all the assays. Cisplatin, taxol, or doxorubicin should be included, even in the in vivo assay. The authors are encouraged to perform the experiments with positive control. 

Is there a pharmacological relevance for evaluating a compound at 0.5 and 1 micromolar? This only represents double the initial concentration.

In vivo assay. 

a) The growth of the tumors seems not significant between the control group and the treatment. Therefore, it is necessary to include a positive control to guarantee the quality of the experiment. 

b) Why did the authors only test one single dose? 

c) This experiment should be repeated, probably the number of cells inoculated in each mouse was not adequate. Images of tumor histology are not shown. 

Round 2

Reviewer 2 Report

Cell cycle distribution

A positive control is also necessary for this assay.

In vivo assay.

The growth of the tumors seems not significant between the control group and the treatment. Therefore, it is necessary to include a positive control to guarantee the quality of the experiment. I understand the number of animals should be reduced. However, in your manuscript, the size of the tumors in the vehicle and treatment group is not homogenous. This pattern could be attributed to a mistake in the subcutaneous administration of cancer cells to induce tumors. Positive control is necessary as an indicator of quality.
